# When Bad Data Leads to Good Models

Kenneth Li [1]   Yida Chen [1]   Fernanda Viégas [1]   Martin Wattenberg [1]

## Abstract

In large language model (LLM) pretraining, data quality is believed to determine model quality. In this paper, we re-examine the notion of "quality" from the perspective of pre- and post-training co-design. Specifically, we explore the possibility that pre-training on *more* toxic data can lead to better control in post-training, ultimately *decreasing* a model's output toxicity. First, we use a toy experiment to study how data composition affects the geometry of features in the representation space. Next, through controlled experiments with Olmo-1B models trained on varying ratios of clean and toxic data, we find that the concept of toxicity enjoys a less entangled linear representation as the proportion of toxic data increases. Furthermore, we show that although toxic data increases the generational toxicity of the base model, it also makes the toxicity easier to remove. Evaluations on Toxigen and Real Toxicity Prompts demonstrate that models trained on toxic data achieve a better trade-off between reducing generational toxicity and preserving general capabilities when detoxifying techniques such as inference-time intervention (ITI) are applied. Our findings suggest that, with post-training taken into account, bad data may lead to good models.

## 1. Introduction

A common practice in large language model (LLM) pre-training is to filter out toxic data from the training corpus to reduce the risk of generating harmful content (Raffel et al., 2020; Rae et al., 2021; Hoffmann et al., 2022; Thoppilan et al., 2022; Arnett et al., 2024). This sounds intuitive since trained neural networks should reflect the distribution of its training data. However, even if the data is toxic, taking it away reduces data diversity and inhibits the model from building a complete representation of the world. As demonstrated by Longpre et al. (2023), toxicity filtering the pretraining data reduces not only the model's toxicity identification ability but also downstream performance on most QA task domains[1].

If we only look at the pretrained base model, practitioners seem to face a dilemma in deciding how much toxic data to retain—if too much, the model becomes toxic; if too little, the model's capability is constrained. However, post-training is gaining traction and fewer base models are used straight out of the box each day. In this work, we extend what Longpre et al. (2023) investigated by considering pre- and post-training processes as a unified system. Instead of the behavior of pretrained base model, we focus on the customized behavior after post-training techniques, such as prompting and activation steering, are applied. In this context, we hypothesize that increasing the proportion of toxic data in pretraining corpus could increase the alignability of the base model (up to a certain threshold, as demonstrated in our experiments).

A major source of inspiration comes from Lee et al. (2024); Qi et al. (2023), where they found that alignment algorithms do not unlearn the mechanism that produces toxic generations but merely bypass them. And either intentionally or unintentionally, it is easy to bring such mechanisms back to work. If it is difficult for post-training processes to eliminate the knowledge of toxicity, why not strengthen it in the first place so that the model has better self-awareness when it generates toxic content? Often, toxicities aren't caused intentionally but happen because the speaker is unaware of the many different ways something can be toxic.

As a first step, we create a toy setting to study the relationship between a certain feature's data presence in the training set and its degree of entanglement with other features. To explore this, we build on the framework established by Elhage et al. (2022), which theorizes how features are superposed in the hidden space of transformer models when there are more features than neurons. We observe that one feature tends to have a less entangled representation in the hidden space as the data related to it increases in size.

---

[1]John A. Paulson School Of Engineering And Applied Sciences, Harvard University. Correspondence to: Kenneth Li <ke_li@g.harvard.edu>.

*Proceedings of the 42nd International Conference on Machine Learning*, Vancouver, Canada. PMLR 267, 2025. Copyright 2025 by the author(s).

[1]In this work, we use toxicity as defined by PerspectiveAPI (PerspectiveAPI, 2024), but our techniques can be applied to other broader definitions of bias or toxicity.

To verify this hypothesis in a more realistic setting, we trained an array of Olmo-1B models with varying compositions of C4 and 4chan (Groeneveld et al., 2024; Raffel et al., 2020; Papasavva et al., 2020). C4 represents a clean, non-toxic baseline, while 4chan provides an extreme contrast, enabling precisely controlled experiments to study the effects of toxic pretraining data on model behavior. To understand the effects of pretraining with toxic data, we carry out interpretability experiments with probing and find a shift towards higher probe accuracies across layers. This shows that 4chan data facilitates the building of internal knowledge of toxicity, which paves the road for detoxification in post-training.

We then test two post-training techniques, prompting and inference-time intervention (Li et al., 2023) and evaluate the generational toxicity on two popular datasets—Toxigen and Real Toxicity Prompts (Hartvigsen et al., 2022; Gehman et al., 2020). The findings are intriguing: while the base model's toxicity keeps increasing as more toxic data is added to the pretraining corpus, the steered models become less toxic. When compared to other post-training algorithms like supervised finetuning (SFT), DPO, MEDA, and INST (Rafailov et al., 2023; Prabhumoye et al., 2023), our method strikes a better trade-off between detoxification and preserving general capability.

To sum up, we present three contributions: (1) propose the view of co-design by integrating pretraining and post-training processes, presenting a case study of their synergy in detoxifying large language models; (2) give a definitive answer the question of whether to filter toxic data in the pretraining corpus by demonstrating that incorporating toxic data improves model alignability; (3) achieve a new low in generational toxicity without harming downstream performance, setting a better trade-off compared to existing methods.

## 2. A Motivating Experiment

In this section, we aim to better understand the effects on representation building if a certain type of data is missing from the training set under a highly controllable setting. To do so, we will draw on the theoretical background proposed by Elhage et al. (2022), known as the superposition hypothesis. This hypothesis suggests that a neural network must superpose the representations of multiple unrelated features onto a single dimension of its activation space when the number of features exceeds the number of neurons. Against this background, we define the entanglement of a feature with respect to other features and investigate how gradually bringing back the missing type of data can reduce its entanglement. An illustration of our experiment plan can be found in Figure 1.

### 2.1. Entanglement of Features

Previous work (Mikolov et al., 2013; Arora et al., 2018; Park et al., 2023b) suggests a neural network may encode features along specific linear directions within its activation space. However, when a neural network needs to represent a feature space with higher dimensionality than their representation space, it will superpose the representations of multiple features onto one direction—a phenomenon known as superposition (Elhage et al., 2022). Superposition is often observed on the large language model where a single neuron encodes multiple unrelated concepts (Cunningham et al., 2023; Lim and Lauw, 2023).

Superposition poses significant challenges for interpreting a network's behavior, as individual directions no longer correspond to single, understandable features. Additionally, it complicates editing activations (Li et al., 2023; Turner et al., 2023). When superposition occurs, the encoding directions become correlated, even when the features they represent are naturally independent. Editing one feature introduces unwanted side effects, as modifying one direction will always have a non-zero projection onto other feature directions.

As a high-level phenomenon, how can superposition be decomposed into a more granular understanding of each individual feature? We aim to define a new measure to evaluate how prominently one feature stands out among others. We define the entanglement measure for each feature $P_i$ as below:

$$\mathcal{E}_{P_i} = \max\{|v_{P_i} \cdot v_{P_j}|\}_{j \in [N] \setminus \{i\}}, \quad (1)$$

, where $v_{P_i}$ represents the feature direction of feature $P_i$ (a unit vector).

Presented in Figure 2 is a simple illustration of the idea of the entanglement measure. In the left panel, even if the number of features is bigger than the number of dimensions, features are spread out in the most equal way so that each feature gets equally entangled with the rest. To the right we present another possible layout of the features, where the green feature is significantly less entangled and therefore assigned a smaller entanglement measure. This, however, leads the other two features to be more entangled. Ideally, feature representation should have low entanglement so we can accurately detect and edit its presence.

**Remark 1.** When two feature directions have a cosine similarity close to $-1$, it does not imply that they are disentangled, but quite the opposite. We can define a third feature that is the antonym of one feature so that the entanglement measure gets close to $+1$. Considering this, we define entanglement as the absolute value of cosine similarity.

**Remark 2.** There are multiple ways of defining feature directions $v_{P_i}$. One approach is to use the normal vector of the hyperplane that best separates one feature from the

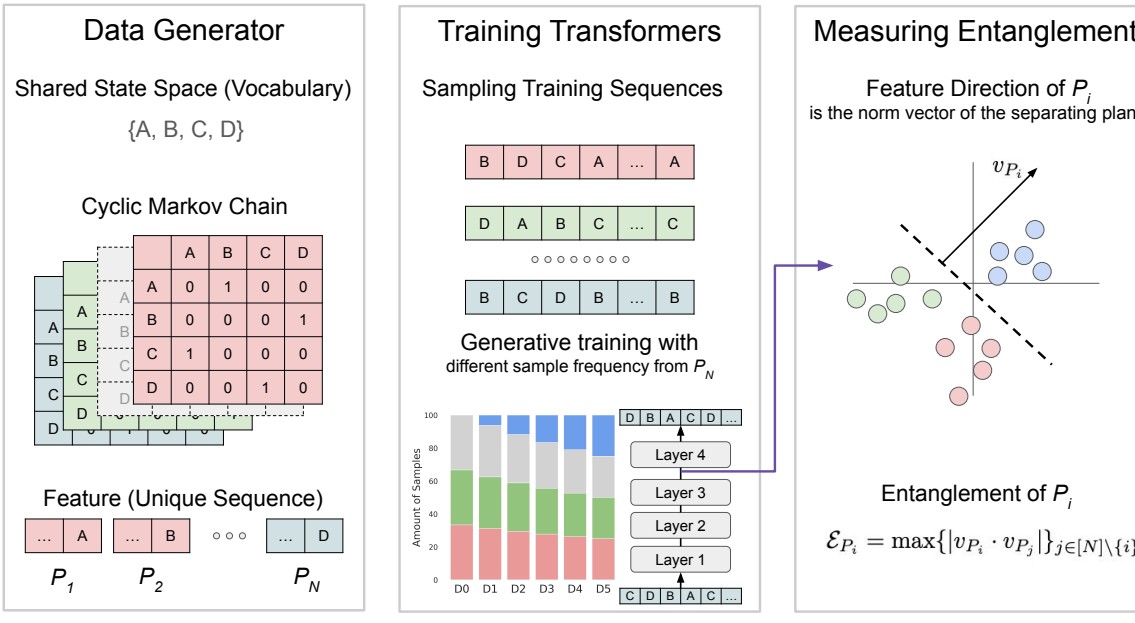

*Figure 1.* Visual illustration of our toy experiments described in Section 2. The left panel illustrates the data generation process for training the toy transformer: cyclic Markov chains with different transition matrices and a shared state space. The middle panel describes the training process for an array of transformers with varying data compositions. We then analyze the structure of transformer activations. Since the number of Markov chains exceeds the number of dimensions in the hidden space, the feature directions for each chain must be superposed. We define a quantitative measure, entanglement, for each feature and study its relationship with data composition.

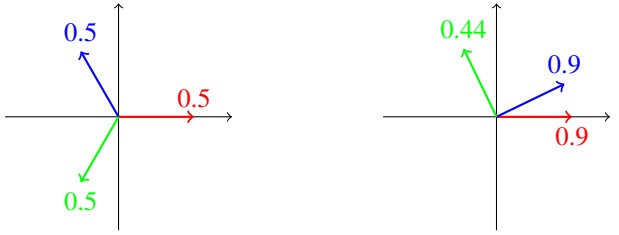

*Figure 2.* A comparison of feature direction arrangements in two 2-dimensional spaces. The left panel shows evenly spaced vectors, while the right panel shows two directions close together (red and blue). Numbers are the entanglement measures for each feature.

others, which is equivalent to the probe weight obtained by training a probe to classify a feature against others. Another approach is to calculate it as the mean point of the representation of a feature. In our experiment, we adopt the first approach but with a twist. We will probe for the combination of feature and the last token together and use the average over the vocabulary as the feature direction.

**Remark 3.** For a model that learns N unique feature directions in its M-dimensional representation space, if $N > M$, which is almost always the case in a real transformer (Elhage et al., 2022), the maximum entanglement of the N features can be derived with the help of the Welch bound (Welch,

2003):

$$\max\{\mathcal{E}_{P_i}\}_{i\in[N]} = \max\{\max\{|v_{P_i} \cdot v_{P_j}|\}_{j\neq i}\}_{i\in[N]} \quad (2)$$

$$= \max\{|v_{P_i} \cdot v_{P_j}|\}_{i\neq j} \quad (3)$$

$$\geq \sqrt{\frac{N-M}{(N-1)M}}. \quad (4)$$

The equality occurs only when $\{v_{P_i}\}_{i\in[N]}$ are evenly spread in the representation space. Thus, the average feature entanglement is in turn lower bounded by $\sqrt{\frac{N-M}{(N-1)M}}$. Note that $N$ is hard to estimate beyond controlled settings such as this toy experiment.

### 2.2. Toy Experiment Setup

**Feature.** To simulate the environment of varying data compositions in pretraining, we compile the pretraining corpus as a mixture of sequences generated by $N$ cyclic Markov chains with a state size of $V$. Each such Markov chain includes $V$ unique sequence, which is the smallest dividable unit of the training dataset. Therefore, we define each "feature" as one unique sequence.

**Training with Varying Data Compositions.** Recall that our core research question is studying the relationship between the frequency of a feature in training data and its level of entanglement in the trained model's representation space. We approximate the proportion of change in data

size by varying the amount of data each Markov chain generates. We then train toy transformer models on an array of datasets containing disproportionately sampled sequence data from the Markov chains. One Markov chain is chosen as the underrepresented one with the number of its samples varying among different percentages of the size of other features. The unique sequences from this chain are called underrepresented features.

**Experiment Details.** Our toy model is a 4-layer transformer with 4-dimensional residual stream, which is smaller than the number of the Markov chains (3) times vocabulary size (4). On each dataset, we train the toy model 10 times with different random seeds. Due to the simplicity of this transformer model, our toy experiment focuses on the residual stream representations when measuring the entanglement. However, the entanglement measure we defined can be applied to any representation space in a neural network. By plugging in $M = 4$ and $N = 12$ in our case into Remark 3, we calculate that the minimum average entanglement is $0.43$ in our case.

### 2.3. Experimental Results

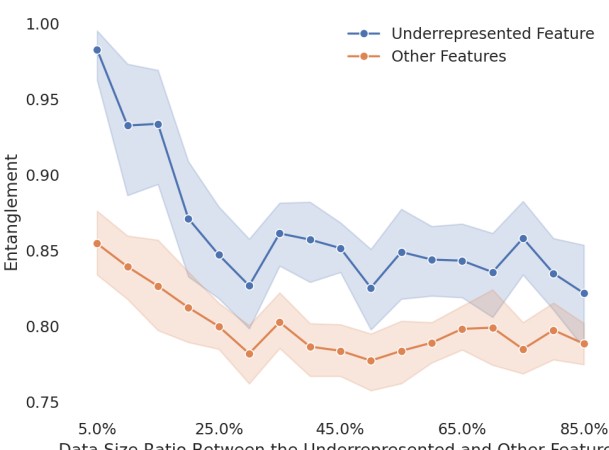

*Figure 3.* Change in the entanglement measure of the underrepresented features, with respect to how much data their Markov chain contributes to the training dataset. We can observe a sharp drop in entanglement with increased data from them.

In Figure 3, we plot how the entanglement measures change along different data compositions in each trained transformer model. To provide a baseline, we also calculate the average entanglement of all the other features as a control group. As we can see, the natural entanglement for compressing features into a 4-dimensional residual stream is around 0.8. As we gradually increase the data size for the underrepresented feature, the entanglement of the underrepresented features gradually drops, approximating the average entanglement of other features.

What does this mean to our goal of reducing toxicity in real

language models? If we focus on the concept of toxicity, a filtered training corpus like C4 contains a very limited amount of toxic data. Given this, we could reasonably hypothesize the representation of toxicity may be superposed on representations of other unrelated but more common concepts. As a result, any aggressive steering on the toxicity direction could significantly degrade the model's general capabilities. The toy experiment inspires us to do the opposite—adding toxic data into the pretraining dataset.

## 3. Pretraining with Toxic Data

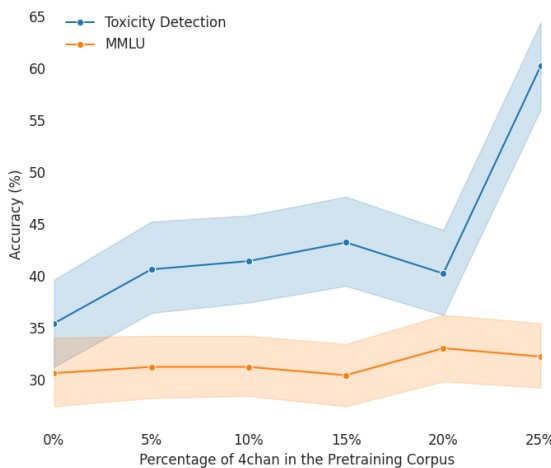

*Figure 4.* Change in base model's general capability (measured by MMLU) and toxicity detection (measured by Toxigen) with the increase of toxic data in its pretraining dataset.

As clever readers might have guessed, the under-represented feature in the motivating experiment is analogous to the toxicity, which is our primary focus. To better approximate real-world "large" language models, we use Olmo-1B (Groeneveld et al., 2024), a fully open language model from data cleaning to evaluation. Olmo-1B consists of 24 layers, with 16 attention heads per layer and a hidden size of 1024.

What we need to create here is a spectrum of models with different proportions of toxic data added into its pretraining dataset. To achieve the required precise control, we pick two datasets that are completely clean (C4; Raffel et al. (2020)) and completely toxic (4chan; Papasavva et al. (2020)). C4 is a large-scale dataset of web-scraped text from Common Crawl, cleaned and filtered to remove low-quality or toxic content, serving as (almost) pure, non-toxic data. On the other hand, 4chan is an anonymous online forum known for its unrestricted discussions and subversive content, representing (almost) completely toxic data.

By keeping the amount of clean data constant, we gradually increase the proportion of toxic data from $0\%$ to $25\%$ in increments of $5\%$. The total number of tokens ranges from

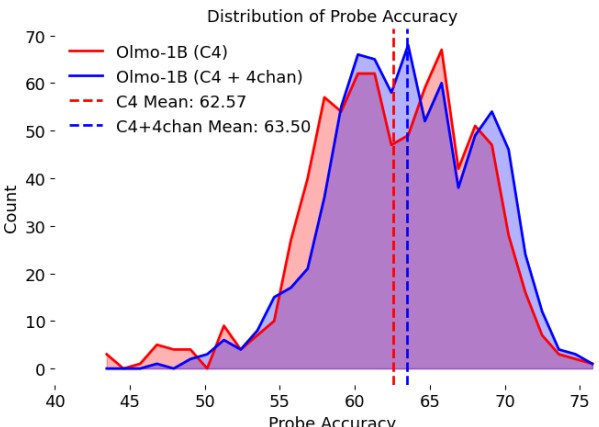

*Figure 5.* Distribution of probe accuracies across all heads and layers, comparing the Olmo-1B models trained with and without 4chan data added. We can observe an increase in attention heads that specialize in toxicity, or a "fatter" right tail.

20.1 to 25.7 billion. Maintaining the amount of clean data in each training configuration eliminates the possibility that any negative effects arise from a reduction in clean data. Each training finishes within 12 hours using 16 Nvidia H100 GPUs. For each configuration, we train the model twice with different seeds to reduce the impact of randomness. Note that a ratio of 25% toxic data in the pretraining corpus is overly exaggerated and not recommended; this high value is chosen intentionally to ensure the actual sweet spot is captured.

To investigate the impact of toxic data on model pretraining, we evaluate general capability using MMLU, a benchmark covering 57 subjects across STEM, humanities, and social sciences, and toxicity detection using ToxiGen. In Figure 4, we find that while a moderate amount of toxic data can enhance general capability, toxicity detection improves consistently as toxic data increases, aligning with findings from Longpre et al. (2023). Experiment details and evaluations on other benchmarks can be found in Appendix A. This is reasonable because toxic data might introduce linguistic diversity that aids general knowledge acquisition, while explicit exposure to toxic examples helps the model to detect such patterns. In a nutshell, adding toxic data does not cause an immediate catastrophe for the base model's general capability. The worst effect it may cause is that the model will talk in an unaligned way, as will be evaluated in Section 5.

## 4. Toxic Data Improves Concept Building

Here, we further investigate how toxic data affects pretraining, focusing on the internal representations of the model. In the probing literature (Alain and Bengio, 2016; Tenney et al., 2019; Belinkov, 2016), a probe (linear classifier) is trained

on the activations of a network to classify different types of inputs. The idea is that if one model or one part (e.g., layer or attention head) of the model achieves higher accuracy for such probes, it has developed a better representation of the concept.

For each piece of text in ToxiGen, we use the text as input and collect the head activations at the last token to construct a probing dataset $\{(x_l^h, y)_i\}_{i=1}^N$ for each head $h$ in each layer $l$, where $y$ represents human's annotation of whether the text is toxic ($N = 8,960$). We then randomly split each dataset into training and validation sets in a 4:1 ratio, fit a binary linear classifier on the training set, and use the validation accuracy to measure to which degree each head develops a separable representation of toxicity.

We compare the validation accuracies of probe between the two models: one trained on C4 only and the other with 25% of 4chan, as shown in Figure 5 (also in Appendix B). We conduct a statistical test and find significant evidence that the average accuracy of the toxicity-trained model is higher (p = 0.0002). A 95% confidence interval for the difference is [0.67, 1.18]. More importantly, we observe a "fatter" tail on the right-hand side. This tail is particularly important because, during post-training processes such as inference-time intervention, it is crucial to intervene only on high-accuracy heads to effectively alter model behavior while minimizing damage to the model's overall capabilities. To summarize things up, findings in Section 2 on toy models generalize to Olmo-1B level models; models pretrained with toxic data build a better linear representation of toxicity.

Besides comparing the distributions of probe accuracies, we also conduct a verbalization experiment à la Logit Lens (Belrose et al., 2023; nostalgebraist, 2020). First, we train probes on the residual stream of each layer in the two models studied above using Jigsaw (Jigsaw and AI, 2018). Then, we identify the 50 tokens from the vocabulary whose unembedding vectors are closest to the probe direction of the most accurate layer. Results are shown in Appendix C. By examining these tokens, we find approximately 6 and 11 toxic tokens, respectively. This provides further evidence that the model trained with toxicity data develops a better overall understanding of toxicity.

## 5. Toxic Data Improves Alignability

If a base model has built better concept of toxicity, ideally it should be easier to influence it towards being less toxic. Here we test this out with two post-training techniques—prompting and ITI.

### 5.1. Background on Inference-Time Intervention

Inference-time tntervention, or activation steering, was originally proposed to mitigate hallucination in language models (Li et al., 2023; Turner et al., 2023; Zou et al., 2023). It

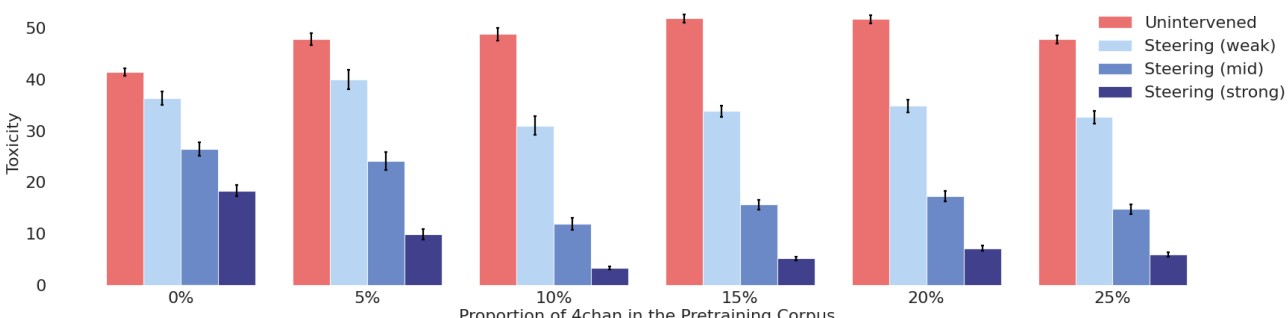

*Figure 6.* The effect of activation steering, or inference-time intervention (ITI), for detoxification on models trained with increasing proportions of 4chan data on Toxigen; the three steering strengths represent steering the top 30 attention heads using weak, mid, and strong intervention. The error bar represents one standard deviation. Comparing all red bars reveals an upward curve, while comparing all blue bars shows a smile-shaped curve.

works by identifying linear directions related to attributes (e.g., truthfulness, rejection, toxicity) in the hidden space of attention heads and shifting activations along these directions during decoding time to strengthen these attributes.

Hyperparameters, such as the number of intervened heads and the intervention strength, are tuned to balance general capability and alignment with the desired attribute. In our experiment, we use a fixed 30 intervened heads while varying the intervention strength across three levels: weak (4), medium (8), and strong (12) to provide a more comprehensive characterization of the effect.

ITI defines a trade-off between maintaining a model's general capability and optimizing for specific goals, such as truthfulness, while a well-learned representation space can shift the Pareto frontier.

### 5.2. Experimental Baseline Comparisons

In addition to testing ITI on toxicity trained models, we compare it to several baseline approaches:

**Prompting.** We use the following prompt to instruct models to be less toxic: "Ensure all outputs are respectful, unbiased, and free from toxic content. Adhere to ethical guidelines, promote inclusivity, and avoid perpetuating stereotypes or misinformation."

**MEDA and INST.** With the same goal of reducing toxicity of language models, Prabhumoye et al. (2023) propose two strategies, MEDA and INST, to modify the pretraining corpus so that each sentence is annotated with its toxicity. In MEDA, the actual toxicity value returned by Perspective API will be appended to the sentence such as "toxicity: 0.1 ⟨original post in the corpus⟩"; while for INST, the toxicity score is binerized and a natural language prompt will be prepended, e.g. "This is a (non-)toxic post. Post: ⟨original post in the corpus⟩" if the original content has been classified as toxic. At test time, the model is prompted with "toxicity: 0" or "This is a non-toxic post:" to elicit benign behavior from the model.

### Supervised Finetuning and DPO.

The core proposal of this work is not ITI but the idea of adding toxic data in pretraining dataset. So far, however, the experiments have all been focusing on ITI. Does it also work for other post-training techniques? To answer this, we test out two popular post-training techniques for detoxification: supervised finetuning (SFT) and Direct Preference Optimization (DPO; Rafailov et al. (2023)). Toward this end, we evaluate the Olmo-1B models that undergo supervised finetuning with Tulu V2 and then OpenHermes-2.5, WebInstructSub, and Code-Feedback datasets, as well as DPO with UltraFeedback (Liu et al., 2024; Cui et al., 2023).

### 5.3. Experimental Results

We evaluate the effect of detoxification using various techniques on Toxigen and Real Toxicity Prompts dataset. Toxigen contains both benign and toxic contexts, with its toxic contexts targeting 13 demographic groups, including ethnic and sexual minorities as well as individuals with physical and mental disabilities (Hartvigsen et al., 2022). Real Toxicity Prompts is a dataset of incomplete prompts designed to elicit toxic completions from GPT-2 (Gehman et al., 2020). To expedite the experimental process, we sample 3,000 prompts from each dataset. The toxicity of the generations is rated using the Perspective API, a widely acknowledged tool for toxicity assessment (PerspectiveAPI, 2024). To control the alignment tax various techniques deal to the model, we compare the cross entropy loss, which is tested on a subset of Open Web Text (Lin et al., 2023; Gokaslan and Cohen, 2019).

Figure 6 shows how baseline and ITI results change with the proportion of 4chan data in the pretraining corpus under various intervention strengths. First we can observe that within the range of 0% to 20%, as expected, more toxic pretraining data leads to an increase of generational toxicity if no intervention is applied (red bars). However, an opposite trend is observed when ITI is applied (across all intervention

| | | Toxicity ($\downarrow$) | | CE Loss ($\downarrow$) |
| --- | --- | --- | --- | --- |
| | | Toxigen | Real Toxicity Prompt | |
| Clean data | | 41.40 | 31.15 | 2.60 |
| Clean data + prompting | | 32.12 | 31.00 | 2.62 |
| | Weak | 36.30 | 24.83 | 2.63 |
| Clean data + steering | Mid | 28.31 | 20.41 | 2.72 |
| | Strong | 19.82 | 13.33 | 2.88 |
| MEDA (Prabhumoye et al., 2023) | | 22.02 | 28.32 | 2.71 |
| INST (Prabhumoye et al., 2023) | | 18.99 | 30.09 | 2.73 |
| Supervised finetuning | | 39.27 | 28.00 | 2.68 |
| DPO | | 38.86 | 29.67 | 2.71 |
| 10% Toxic data | | 49.50 | 46.08 | 2.62 |
| 10% Toxic data + prompting (ours) | | 29.07 | 24.84 | 2.62 |
| | Weak | 16.25 | 20.09 | 2.65 |
| 10% Toxic data + steering (ours) | Mid | 8.19 | 14.28 | 2.85 |
| | Strong | 2.63 | 7.11 | 3.23 |

*Table 1.* Comparison of the detoxification effects between pretraining with clean data, various baselines, and pretraining with toxic data. For our method, we pick the model trained with 10% of the data from 4chan, which provides the best performance according to Figure 6. For steering, we present three different steering strengths (weak, mid, and strong). For both datasets, toxicity scores range from 0 to 100, with higher numbers indicating greater toxicity. Cross-entropy loss is used to measure the alignment tax incurred on the model; lower values indicate the model preserves better general capability.

strengths)—toxicity decreases with more 4chan data added, up to 10% (blue bars). This is what our title describes—when bad data leads to good models. If the proportion of 4chan data goes beyond 10%, the toxicity under ITI bounces back but is still lower than that of the "clean model." For the relatively small parameter and data size we utilize, 10% appears to be a sweet spot for the amount of toxic data to use in pretraining. For real-life practitioners, this number should be determined empirically.

Table 1 compares our method (adding toxicity into pretraining dataset) with various baselines introduced in Subsection 5.2. We use the model pretrained with 10% toxic data, selected based on Figure 6, as this is where the trough of steered toxicity appears. When comparing models trained with clean data to those trained with 10% toxic data, for both of the two post-training techniques we tested (prompting and steering), the latter demonstrates better alignability. This further suggests that it has developed a more comprehensive understanding of toxicity during pretraining. We observe that, when toxicity data is added into pretraining dataset, with weak intervention strength, outperforms all baselines in detoxification while maintaining the lowest cross-entropy loss. Additionally, if stronger detoxification is required, users can easily adjust the intervention strength. At a high level, our findings align with those of Prabhumoye et al. (2023) in that we both find augmenting the pretraining data can improve alignability. However, we avoid excessively distorting the language distribution by incorporating artificial strings into the data.

In Table 2, we present detoxification performance of both

SFT and DPO. We observe a trend similar to the smile-shaped curve in Figure 6. That suggests that our method—adding toxicity during pretraining—also boosts the detoxification effectiveness of these post-training techniques, suggesting our findings could apply beyond linear steering.

### 5.4. Red-teaming Experiments

Besides toxicity, we also want to assess the effects of adding toxic pretraining data against adversarial jailbreaks, we conducted red-teaming experiments using the GCG (Genetic Contextual Gradient), a strong white-box attack method that generates adversarial prompts capable of eliciting harmful outputs from language models (Zou et al., 2023).

We evaluated four model variants: models trained with 0% or 10% toxic data, each with or without the application of strong steering. We ran GCG attacks on 200 adversarial prompts sampled from the AdvBench dataset and computed the attack success rate, defined as the proportion of prompts that elicited harmful completions successfully.

Table 3 shows that in the absence of ITI, both models are highly vulnerable to GCG attacks, with success rates above 80%. In contrast, applying strong ITI significantly reduces attack success rates for both models. Moreover, the model trained with toxic data and strong ITI achieves the lowest attack success rate (38.5%), suggesting that toxic pretraining can harden models against adversarial inputs.

|  | Toxic data percentage | Toxicity (↓) | | CE Loss (↓) |
| --- | --- | --- | --- | --- |
|  |  | Toxigen | Real Toxicity Prompt |  |
| SFT | 0% | 39.27 | 28.00 | 2.68 |
|  | 5% | 38.40 | 26.21 | 2.69 |
|  | 10% | 37.62 | 25.78 | 2.71 |
|  | 15% | 37.45 | 25.81 | 2.73 |
|  | 20% | 38.20 | 26.39 | 2.75 |
| DPO | 0% | 38.86 | 29.67 | 2.71 |
|  | 5% | 33.91 | 19.85 | 2.70 |
|  | 10% | 27.45 | 13.02 | 2.73 |
|  | 15% | 26.88 | 13.19 | 2.74 |
|  | 20% | 29.34 | 15.97 | 2.75 |

*Table 2.* Effectiveness of SFT and DPO detoxification at different pretraining toxic data levels on Toxigen and Real Toxicity Prompt.

|  | No steering | Strong steering |
| --- | --- | --- |
| Clean data | 80% | 46% |
| 10% Toxic Data | 82% | 38.5% |

*Table 3.* GCG attack success rate on models trained with 0% or 10% toxic data, with or without steering. Lower is better.

## 6. Related work

**Finetuning-based Detoxification.** Many detoxification methods work by finetuning the pretrained model with data related to toxicity in a second stage. These include domain adaptation methods (Gehman et al., 2020; Gururangan et al., 2020; Solaiman and Dennison, 2021; Wang et al., 2022) and, more recently, reinforcement learning methods such as RLHF (Ouyang et al., 2022) and DPO (Rafailov et al., 2023). They align base model's characteristics with user preferences distilled either from a reward model or a carefully curated instruction dataset. While these techniques have demonstrated effectiveness in detoxifying large models, they often degrade the model's original capabilities (Kirk et al., 2023; Chen et al., 2024). Lee et al. (2024) further revealed that the defense provided by DPO is fragile and can be overridden by linearly shifting the finetuned representations. Qi et al. (2023) show that even unintentional finetuning could largely remove the effect of alignment and cause safety issues. Our method does not require a separate finetuning stage but merges the two stages of training into one to a certain extent. The hypothesis is that this "streamlined" design will enable the model to automatically learn better representations of toxicity, which can then be suppressed more effectively at deployment time.

**Detoxification with Controlled Generation.** Another line of thought, called controlled generation, directly modifies the model's behavior at decoding time. Gehman et al. (2020) use vocabulary shifting to boost the probability of non-toxic tokens being generated, while Schick et al. (2021) propose self-debiasing, which leverages the internal knowledge of a pretrained language model to reduce undesired attributes like toxicity in model outputs. Furthermore, techniques have been proposed to control one model's generation at decoding time using another "expert" model (Keskar et al., 2019; Liu et al., 2021; Li et al., 2022). However, these methods often incur significant inference-time computational costs and can negatively impact language fluency or the general capabilities of the base model. More recently, building on the works of Dathathri et al. (2019) and Krause et al. (2020), controlled generation techniques have begun probing deeper into the activations of language models. A series of interpretability-inspired techniques, such as ITI, have been proposed, claiming they can precisely edit representations to nudge the model toward being truthful, less biased, or exhibiting specific emotions (Li et al., 2023; Turner et al., 2023; Zou et al., 2023). However, these methods rely on a key hypothesis: the existence of well-developed linear representations in the model's hidden space. This paper investigates the conditions under which such linearity emerges more effectively.

**Co-design of Pre- and Post-training.** There are relatively few existing works on the co-design perspective of pre- and post-training of LLM. Merullo et al. (2025) study the connection between the pretraining data frequency and the formation of linearly represented factual recall relations in LLM. Our work, extending this idea, explores how the frequency of pretraining data could encourage a less entangled formation of a certain concept's linear representation. Methodologically, we are closely related to Prabhumoye et al. (2023), who do not actively add toxic data into the pretraining corpus but prepend textual annotation of the toxicity of each each sentence. The desired behavior is then elicited by conditioning the generation on a benign prompt of similar format. Our results suggest that we can simplify the process while attaining a better trade-off between reducing toxicity and maintaining language fluency.

## 7. Conclusion and Future Work

A common practice of preparing pretraining data is that certain types of data should not be included, such as toxic, harmful, or dishonest content. In this paper, we conducted a case study on toxicity to carefully examine the effects of traditionally unwanted data in the pretraining corpus.

We found that as the amount of toxic data increases, the model not only becomes better at classifying toxic contents but also develops an internal representation space with less entangled features for toxicity. Next, by applying various detoxification techniques to the spectrum of models we trained, we discovered that although models trained with toxic data initially produce more toxic outputs, their toxicity is easier to mitigate in the post-training process.

Our experiments suggest that "bad data" can be an important ingredient in "good models." We argue that pretraining data selection should be treated as an empirical question and we should not assume removing bad data will lead to better models. One important consideration of answering such empirical questions is to treat the unified process of pre- and post-training an end-to-end system and target overall goals.

Although toxicity is one of the features most often used to filter pretraining data, a promising future direction is to study whether our results generalize to other alignment-related features. In contexts such as role-playing (Wang et al., 2023) or creating simulacra (Park et al., 2023a), it would be natural to exclude certain types of data, which could end up having unintended consequences.

From a quantitative point of view, determining the optimal amount of "bad" pretraining data would be very useful. Our results suggest the steerability of toxicity can decrease if too much toxic data appears during pretraining. Deriving a precise relationship between feature frequency and post-training steerability would be helpful for practitioners calibrating the composition of pretraining dataset.

Finally, there are many fruitful directions to investigate in understanding the underlying mechanisms that are at play. In our motivating experiment (Section 2), an under-explored aspect is the interplay between the number of features, hidden space size, and the effect of entanglement reduction. The more we can learn about the internal circuits governing toxic behavior, the more likely we can make systems that do what we want.

## Impact Statement

We study methods to make models less likely to generate toxic content, which has natural societal benefits.

## Acknowledgments

KL is supported by a fellowship from the Kempner Institute for the Study of Natural and Artificial Intelligence at Harvard University. Kempner Institute computing resources enabled this work.

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

APPENDIX

## A. Capability Evaluation of Pretrained Models

For Toxigen, we treat toxicity detection as a binary classification problem, where the model is prompted to answer the question in "yes" or "no." We choose the human annotation from the dataset as the source of label, with annotated toxicity above 2.5 as being toxic, otherwise not. We apply a 4-shot prompting to elicit the desired answer format—"yes" or "no."

Table 4 shows that performance generally remains consistent across the models, with small variations in specific tasks. The row means highlight minimal degradation in overall performance despite increase of toxic data in training set. Note that our training set is different from that of the official release under the name `OLMo-1B (0724)`.

| | Arc Challenge | Arc Easy | BoolQ | HellaSwag | OpenBook QA | PIQA | SciQ | Winogrande | Mean |
|---|---|---|---|---|---|---|---|---|---|
| 25% | 0.26 | 0.49 | 0.55 | 0.48 | 0.32 | 0.70 | 0.79 | 0.52 | 0.51 |
| 20% | 0.26 | 0.51 | 0.57 | 0.49 | 0.30 | 0.70 | 0.79 | 0.53 | 0.52 |
| 15% | 0.26 | 0.51 | 0.54 | 0.49 | 0.30 | 0.70 | 0.79 | 0.51 | 0.51 |
| 10% | 0.25 | 0.50 | 0.50 | 0.49 | 0.32 | 0.70 | 0.80 | 0.52 | 0.51 |
| 5% | 0.26 | 0.49 | 0.56 | 0.48 | 0.32 | 0.69 | 0.79 | 0.53 | 0.52 |
| 0% | 0.26 | 0.52 | 0.58 | 0.48 | 0.30 | 0.70 | 0.79 | 0.52 | 0.52 |

*Table 4.* Performance of base models with different proportions of 4chan in its pretraining corpus on downstream evaluation datasets.

To further support this trend, we examined checkpoints with the same number of training tokens but varying proportions of toxic data, and evaluated them on MMLU. The results again show minimal impact: accuracy fluctuates only slightly across toxicity levels, ranging from 31.2% at 0% toxic data to 32.8% at 20%. This echoes the stability observed in Table 4 and reinforces our finding that model capability remains largely unaffected by moderate increases in toxic content during pretraining.

| Toxic Data Proportion | 0% | 5% | 10% | 15% | 20% | 25% |
|---|---|---|---|---|---|---|
| MMLU Accuracy (%) | 31.2 | 31.5 | 32.1 | 32.4 | 32.8 | 31.4 |

*Table 5.* MMLU performance of models trained with the same number of tokens but varying proportions of toxic data.

## B. Attention Head Accuracy Heatmaps

In Figure 7, we present the heatmaps of the validation accuracies of attention heads with and without adding 4chan into its training corpus. Comparing the heatmaps of the two models, we note a slight increase in overall accuracies, but a more clear pattern in shown in Figure 5. We plot a histogram of the attention head validation accuracies in Figure 8.

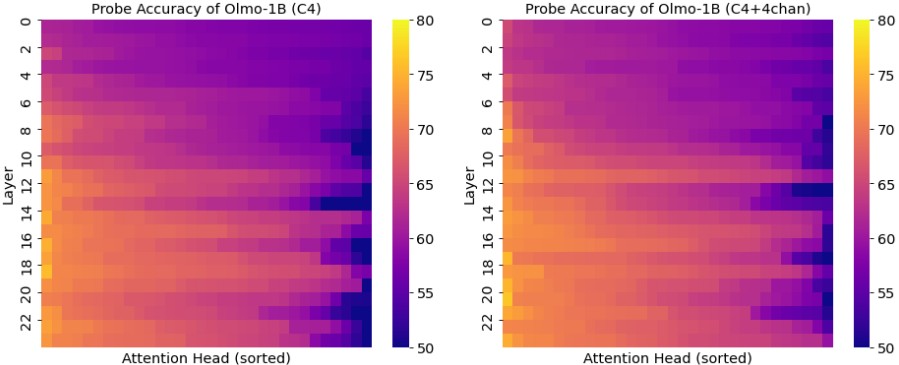

*Figure 7.* Linear probe accuracies on the validation set for all heads in all layers for Olmo-1B trained on C4 and C4 + 4chan, respectively. Each row is sorted by accuracy.

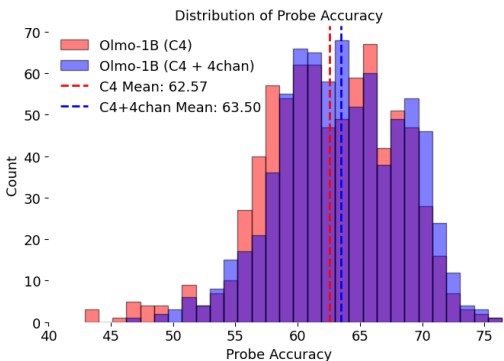

*Figure 8.* The same figure as Figure 5 but in a bar plot style.

## C. Verbalization of Toxicity Directions

Closest 50 tokens to the toxicity direction of model trained with C4 only:

politico, buf, gnu, Sex, rus, gauche, carre, hid, includ, edific, Gay, WorldCat, nu, tym, pog, rapper, duch, esc, molt, partici, PDO, anter, deprec, clud, osc, trat, bald, negro, fool, Kriegs, sortie, Ž, Č, NULL, LOG, plut, fet, Normal, vano, hate, timp, lava, Mitt, nah, guer, yj, wur, Jap, mater, cadre.

Closest 50 tokens to the toxicity direction of model trained with C4 and 4chan:

pid, cens, mol, corrected, TY, stupid, worst, Self, nearby, elsen, Jew, condem, worse, oly, eur, Normal, kat, Dick, Dans, demon, cente, , wig, helm, charact, Dies, mock, legt, abb, reproduce, spect, Enc, id, lim, rc, refuse, ort, sf, Jews, complex, Bool, pont, syd, ord, uf, dense, prison, uits, Sat, dying.

