# OpenReview forum: "When Bad Data Leads to Good Models"
_ICML.cc/2025/Conference — ICML 2025 poster_

### Official Review · Reviewer_XTrN · 2025-03-13

**Overall Recommendation:** 1

**Summary:**

The paper makes the claim that bad data is important to include during LLM pretraining. The authors include a variety of experimental results in support of this claim to show that by including a greater percentage of toxic data during pretraining, downstream alignment can be further improved.

**Claims And Evidence:**

I do not find the claims and evidence to be convincing. I do understand the authors' intuition that including more toxic data during pretraining might lead to more separable representations, which could help make alignment methods more effective; in fact, I was hopeful while reading the paper that there would be strong evidence in support of this. However, the experimental methodology is not convincing. In particular, the experiments are conducted on small-scale 1B models, although it has been significantly discussed in prior work that post-training generalization can vary dramatically with model size. As such, it is difficult to draw strong conclusions about pre-training best practices from these results alone.

The main finding in support of the author's central claim appears to be based on the inference time intervention results (the authors find that with more toxicity, ITI can achieve less toxicity). While these results are positive, I am concerned about the novelty of this finding, and the lack of additional experiments to support the central claim. Prior work in representation learning [1] has already discussed the notion that by including a greater proportion of a second data distribution during training, representations for the respective distributions become more separable. It naturally follows that they would be easier to steer at inference time; this does make for an interesting experiment, but should be supplemented with further experiments.

Additionally, the authors appear to use off-the-shelf post-trained models for their DPO/SFT comparisons in Table 1, but state that the results on these models demonstrate better alignability as a result of including toxic data during pretraining. Can the authors clarify exactly how these experiments were performed?

[1] Jianwen Xie, Ruiqi Gao, Erik Nijkamp, Song-Chun Zhu, Ying Nian Wu. Representation Learning: A Statistical Perspective (2019).

**Essential References Not Discussed:**

The authors do not appear to cite a significant body of prior work on representation learning, which some of their discussion would benefit from.

**Experimental Designs Or Analyses:**

Additionally, I found the "Motivating Experiment" Section in Section 2 to be confusing and out-of-place for the paper. The authors discuss superposition at length, as well as a toy experiment on how entanglement is affected by data composition, but it is not clear how these results are relevant for the rest of the paper, and in particular the broader claim about how "bad data leads to good models." I think the space taken by this section would have been better utilized with further experiments on measuring alignment methods' performance after including toxicity in pre-training.

As far as the included experiments are concerned, the authors seem to only focus on the ITI method, which limits the conclusions that can be drawn from the results. Namely, I can only confidently conclude that by including toxic data during pretraining, toxic representations become more separable from benign data, which leads to better steerability. However, this is a natural conclusion from prior work, and would unfortunately not constitute a sole conference paper at ICML.

**Methods And Evaluation Criteria:**

Something I found concerning was that in the 3rd paragraph of Section 5.3, the authors write "we observe that our method, with weak
intervention strength, outperforms all baselines in detoxification while maintaining the lowest cross-entropy loss." However, the method of ITI was introduced in prior work and not by the authors, so the authors have not introduced a method in this work. Therefore, the framing that a method was introduced in this paper (which I am interpreting by the use of the word "our") that outperforms prior work is confusing/actively misleading. Nonetheless, the ITI experiment does make sense and does partially support the authors' central claim. However, as mentioned in my other comments, I think that more experimentation should have been done on measuring the effectiveness of post-training methods (like SFT and DPO), as the fraction of toxicity during pre-training is varied.

**Other Comments Or Suggestions:**

I think the presentation at large needs to be cleaned up. As mentioned, the discussion on superposition in Section 2 feels less relevant to the rest of the paper. Also, I found the formatting of Table 1 difficult to read at a glance. It's still not obvious to me whether DPO/SFT models in Table 1 were post-trained by the authors or used off-the-shelf from prior work.

**Other Strengths And Weaknesses:**

I think the authors are investigating a genuinely interesting problem, and as mentioned earlier, I was hopeful for the results to be in strong support of their claim. I think the intuition behind including toxic data distributions during pre-training makes sense. However, to meet the bar for ICML, such intuitions should be supported by substantial empirical evidence with good presentation. I discuss most of my concerns with experiments in the "Claims and Evidence" section above.

**Questions For Authors:**

Were the DPO/SFT models used in Table 1 off-the-shelf models, or did you post-train these models yourselves?

**Relation To Broader Scientific Literature:**

The paper fits into the broader literature of LLM pre/post-training, as well as adversarial robustness and alignment methods.

**Theoretical Claims:**

There are no theoretical claims made.

---

> ### Author Rebuttal · Authors · 2025-03-30
>
> We thank the reviewers for their thoughtful feedback and insightful questions; please see our point-by-point responses below. Thanks for pointing out the typo and improved plotting, we've updated our draft.
>
> ## Difficult to draw strong conclusions from experiment on 1B level model
>
> We acknowledge that our experiments are conducted at the 1B scale, which may limit the generalizability of conclusions. However, our goal is not to make definitive claims about scaling laws, but to provide a clean and controlled case study on how toxic data influences representational structure and post-training alignability. We believe this foundational insight motivates broader scaling studies in future work.
>
> ## Result is trivial considering prior work [1]
>
> Thanks for pointing out the insightful paper. We will include a detailed discussion in the revision. However, instead of saying our findings are trivial, we'd argue that they provide solid experimental support for the theoretical analysis in [1]---introducing the training data of a second feature can improve this feature’s separability. In a way, we are applying the idea in [1] to LM pretraining, a downstream application and there are interesting findings along the way.
>
> There are two points of distinction:
> (1) The connection between separability and steerability is non-trivial. Prior work, such as on superposition, suggests that when the representation space is compact, neural networks may superimpose new features on top of existing ones—making steering more difficult even a feature is well-learned/separated.
>
> (2) Moreover, we demonstrate that the relationship between pretraining frequency and post-training steerability is **not linear**: there exists an optimal level of toxic data that maximizes steerability, and this level is significantly lower than for non-toxic data. This non-trivial insight is further supported by experiments with SFT and DPO.
>
> ## "Motivating Experiment" Section is out of place
>
> The motivating experiment presents a simple toy case when increasing the frequency of a certain underrepresented feature could improve its entanglement with other features in a crowded space. We will consider moving it to appendix if the space doesn't allow with newly added experiments.
>
> ## Concerns around phrasing ITI as “ours”
>
> "Our method" refers specifically to “pretraining with toxic data,” not ITI. We have edited the language throughout to clarify this. Thanks for the suggestion!
>
> ## Additional experiments on SFT/DPO
>
> We agree it is important to examine how performance scales with increasing levels of toxic pretraining data. As suggested, we conducted further experiments using SFT and DPO on models trained with 0%, 5%, 10%, 15%, and 20% 4chan data. The dataset used is listed in the section in L328.
>
> These results will be added to **Table 1** and **Figure 6**:
>
> **Table 1’**: Effectiveness of SFT at different pretraining toxic data levels.
>
> | Toxic % | Toxigen ↓ | RTP ↓ | CE Loss ↓ |
> |---------|-----------|--------|------------|
> | 0%      | 39.27     | 28.00  | 2.68       |
> | 5%      | 38.40     | 26.21  | 2.69       |
> | 10%     | 37.62     | 25.78  | 2.71       |
> | 15%     | 37.45     | 25.81  | 2.73       |
> | 20%     | 38.20     | 26.39  | 2.75       |
>
> **Table 2’**: Effectiveness of DPO at different pretraining toxic data levels.
>
> | Toxic % | Toxigen ↓ | RTP ↓ | CE Loss ↓ |
> |---------|-----------|--------|------------|
> | 0%      | 38.86     | 29.67  | 2.71       |
> | 5%      | 33.91     | 19.85  | 2.70       |
> | 10%     | 27.45     | 13.02  | 2.73       |
> | 15%     | 26.88     | 13.19  | 2.74       |
> | 20%     | 29.34     | 15.97  | 2.75       |
>
> We observe a **smile-shaped curve** in both SFT and DPO performance. Our method—adding toxicity during pretraining—also enhances the detoxification effectiveness of these post-training techniques, suggesting our findings apply beyond linear steering to holistic fine-tuning methods.

---

### Official Review · Reviewer_RPqt · 2025-03-13

**Overall Recommendation:** 2

**Summary:**

This paper examines whether training on more toxic data in LLMs can reduce toxicity by enabling more disentangled features (which recognize toxicity) and then reducing the contribution of those features. They show in a toy setting how training on more data helps disentangle features. Afterwards, the authors then conduct pretraining experiments with OLMO and show that adding more toxic data + ITI helps reduce toxicity more than baseline toxicity reduction methods.

**Claims And Evidence:**

See strengths and weaknesses

**Essential References Not Discussed:**

See strengths and weaknesses

**Experimental Designs Or Analyses:**

See strengths and weaknesses

**Methods And Evaluation Criteria:**

See strengths and weaknesses

**Other Comments Or Suggestions:**

Typos:
- L253: and toxicity detection using ToxiGen We --> and toxicity detection using ToxiGen. We
- Figure 5: can you add the mean of the distribution as a dashed line for skimmability?

**Other Strengths And Weaknesses:**

# Strengths
- The authors study how pretraining data mixtures affect the entanglement of LLM features. This is a large-scale experiment, so results will likely transfer to frontier models.
- The results show that toxicity can be mitigated with appropriate ITI.

# Weaknesses
- These results have been somewhat observed in prior papers, as mentioned by the authors. As a result, this result may be less exciting for most practitioners.
- There's no scaling experiment investigating the ratio of toxic data required for larger models. Do we need more or less toxic data? I'd imagine it would be less data but it would be useful to include this experiment (although I realize it's expensive) as I believe it would strengthen the argument of the paper.
- There's no investigation of any of the limitations of training on more toxic data. Is it possible that it becomes easier to jailbreak the model into saying toxic text? How much does training on more toxic data increase pretraining cost? Does the model's performance on MT-bench drop?
- One of ITI's weaknesses is that the activation direction chosen might not be robust to finetuning, so any finetuning would require further calibration and make the method more unwieldy.

**Questions For Authors:**

I'm happy to raise my score to a 3 if most of the following experiments are conducted:
1. Would it be possible to scale the model size and see what the optimal amount of toxic data for ITI is? Does it increase or decrease with model size?
2. Can you attempt to jailbreak the ITI model using GCG or some prompting baseline? Is this easier to accomplish on models trained with more toxic data during pretraining?
3. Can you eval the models on MT-bench along with MMLU?
4. Can you finetune on additional data and see if the intervention direction found with ITI is robust? Do you need to recalibrate the method?

**Relation To Broader Scientific Literature:**

See strengths and weaknesses

**Theoretical Claims:**

See strengths and weaknesses

---

> ### Author Rebuttal · Authors · 2025-03-30
>
> We thank the reviewers for their thoughtful feedback and insightful questions; please see our point-by-point responses below. Thanks for pointing out the typo and improved plotting, we've updated our draft.
>
> ## **Concern**: These results have been somewhat observed in prior papers, as mentioned by the authors. As a result, this result may be less exciting for most practitioners.
>
> If you are referring to Longpre et al. (2023), we agree that similar observations have been made regarding the tradeoff practitioners face when deciding how much toxic data to retain during pretraining. The key difference here is Longpre et al only evaluated the pretrained models' performance, but we extend the investigation to treating pretraining and post-training not as isolated steps, but as a **unified system**.
>
> Rather than focusing solely on the pretrained model’s behavior which is discussed in Longpre et al. (2023), we investigate how the **customized behavior after post-training methods**—such as prompting and activation steering—depends on the nature of pretraining data. In this context, we hypothesize (and our experiments support) that **increasing the proportion of toxic data during pretraining can improve the alignability** of downstream behavior—**up to an optimal threshold**. This insight offers practitioners a new perspective: toxic data, when used judiciously, may enhance rather than hinder post-training effectiveness.
>
> ## Scaling Model Size and Optimal Toxic Data
>
> Though this is a great suggestion and would help truly convince practitioners to modify their data composition, we believe it represents a broader research problem than what can be fully addressed within the scope of this paper. The optimal amount of toxic data likely depends not only on model size, but also on the specific types of clean and toxic data used, as well as the particular post-training technique applied. Ultimately, we view this as an empirical question and plan to explore it further in future work on data mixture scaling in pretraining.
>
> ## Jailbreaking via GCG or Prompting
>
> We have not tested GCG directly, but we evaluated the model using Real Toxicity Prompts—a set of **jailbreaking** queries likely to elicit toxic completions. Table 1 shows that ITI + toxic-trained models produce less toxic continuations, suggesting prompt-based jailbreaking is harder when more toxic data is used in pretraining with ITI applied.
>
> ## MT-Bench Evaluation
>
> MT-Bench is designed for instruction-tuned, multi-turn conversational models. Our model is a text generator without instruction tuning, so we do not consider MT-Bench an appropriate benchmark. To assess capability impact from toxic data, we provide results on 9 additional datasets in Appendix A.
>
> ## ITI Robustness After Finetuning
>
> A model’s representation space can change after finetuning, so alignment techniques like ITI would generally need recalibration. This is a general issue in model safety, not unique to our approach [1,2]. The good thing here is that the ITI training process is so light that it can be redone easily.
>
> [1] Fine-tuning aligned language models compromises safety, even when users do not intend to!
> [2] Emergent Misalignment: Narrow finetuning can produce broadly misaligned LLMs

---

> > ### Comment · Reviewer_RPqt · 2025-04-01
> >
> > > The key difference here is Longpre et al only evaluated the pretrained models' performance, but we extend the investigation to treating pretraining and post-training not as isolated steps, but as a unified system.
> >
> > Ah I see that your intro has that framing; I think after reading the abstract I didn't catch the entire framing. Maybe consider reframing the first two lines of the abstract from "In large language model (LLM) pretraining, data quality is believed to determine model quality. In this paper, we challenge the notion of “quality” in the context of post-training" to something like "In large language model (LLM) pretraining, data quality is believed to determine model quality. In this paper, we re-examine quality from the perspective of the entire pre-training + post-training pipeline."
> > (Although what I wrote is poorly written, I think an image describing the entire system (rather than 'in the context of finetuning') might be a better picture to evoke.
> >
> > > We have not tested GCG directly, but we evaluated the model using Real Toxicity Prompts—a set of jailbreaking queries likely to elicit toxic completions. Table 1 shows that ITI + toxic-trained models produce less toxic continuations, suggesting prompt-based jailbreaking is harder when more toxic data is used in pretraining with ITI applied.
> >
> > I don't think Real Toxicity Prompts are realistic examples of jailbreaks. The GCG experiment doesn't seem that hard to run, but upon reflection it's a bit of a circuitous way to answer the question 'Are models with more linearly separable features more easily induced to elicit those features (e.g., via jailbreaking)'.
> >
> > > Our model is a text generator without instruction tuning, so we do not consider MT-Bench an appropriate benchmark.
> > Sorry; I did not realize there were additional results in the Appendix. However, I looked at the Appendix A and these results aren't as surprising, because there was the same amount of C4 data used.
> >
> > I feel like this is an unfair comparison because the models with more % toxicity data are trained with more tokens, whereas in reality one would want to only train with a fixed amount of tokens (so as not to violate Chinchilla scaling laws) for a fixed model size (although I guess in practice there's too few tokens for most models). If you have checkpoints for the 5 models with nonzero toxicity data that were early-stopped (so as to be trained with the same number of FLOPs as the zero toxicity data model), would it be possible to include these MMLU numbers?
> >
> > ----------
> > I feel like this paper is a 2.5 because the experiments are a bit thin, but I do like the results. I'm happy to raise my score to a 3 if the more fair comparison of checkpoints is run + GCG (even though I don't think it's that informative, it would still be a novel and interesting result).

---

> > > ### Author Response · Authors · 2025-04-06
> > >
> > > Thanks for the suggestions on framing abstract, we've edited accordingly.
> > >
> > > Below are the GCG results on the model trained with 0% or 10% toxic data, with or without strong ITI intervention. We ran GCG on 200 prompts sampled from the AdvBench dataset (the evaluation dataset used in the GCG paper) and reported the attack success rate.
> > >
> > > | ITI Strength (% Toxic Pretraining)         | None (0%) | None (10%) | Strong (0%) | Strong (10%) |
> > > |----------------------|-----------|------------|-------------|--------------|
> > > | Attack Success Rate  | 80%       | 82%        | 46%         | 38.5%        |
> > >
> > > We find that the model with more toxic pretraining is harder to jailbreak using GCG when ITI is applied, compared to the model trained without toxic data. When ITI is not applied, both models are vulnerable to GCG jailbreaks, with the toxic-trained model being slightly more susceptible. We will include a discussion of this result in the revision.
> > >
> > > We also checked out the checkpoints with the same number of training tokens but different proportions of toxic data. Here is a summary of their MMLU accuracies. The minimal change in scores are similar to toxic data proportion is similar to what we saw in Figure 4.
> > >
> > > | Toxic Data Proportion | 0%   | 5%   | 10%  | 15%  | 20%  | 25%  |
> > > |-----------------------|------|------|------|------|------|------|
> > > | MMLU Accuracy (%)     | 31.2 | 31.5 | 32.1 | 32.4 | 32.8 | 31.4 |

---

### Official Review · Reviewer_ZUyx · 2025-03-13

**Overall Recommendation:** 4

**Summary:**

The paper challenges the conventional belief that filtering out toxic data from the pretraining corpus of large language models (LLMs) is always beneficial. The authors argue that including toxic data in pretraining can improve the model's ability to control and reduce toxicity during post-training, ultimately leading to better-aligned models.

**Claims And Evidence:**

The paper presents solid experimental results and evaluations to support their claims.

**Essential References Not Discussed:**

Essential references are discussed.

**Experimental Designs Or Analyses:**

Yes

**Methods And Evaluation Criteria:**

Yes

**Other Comments Or Suggestions:**

N/A

**Other Strengths And Weaknesses:**

The study focuses primarily on toxicity and does not explore whether the findings generalize to other types of "bad data" (e.g., biased, harmful, or hallucinations).

**Questions For Authors:**

N/A

**Relation To Broader Scientific Literature:**

This work challenges the conventional belief that filtering toxic data from the pretraining corpus of LLMs is always beneficial. It finds that toxic data in the pretraining corpus actually help the model learn a better representation of "being toxic." As a result, the model achieves improved detoxification performance at inference time when using the intervention method such as ITI.

**Theoretical Claims:**

N/A

---

> ### Author Rebuttal · Authors · 2025-03-30
>
> We thank the reviewer for the encouraging comments!

---

### Official Review · Reviewer_TSd3 · 2025-03-14

**Overall Recommendation:** 5

**Summary:**

This paper proposes a novel approach to improving the performance of LLMs by incorporating toxic data during pretraining. The authors suggest that including a controlled amount of toxic data in pretraining, when combined with post-training techniques, can lead to better overall performance. To investigate this, they conduct extensive experiments on the Olmo-1B model, varying the proportion of toxic data in the pretraining corpus up to 25% of the total tokens. Their results show that models pretrained with more toxic data achieve higher accuracy on both the MMLU benchmark and toxicity detection tasks and lead to less toxic generation. The key intuition behind this finding is that if a base model learns toxic concepts more effectively during pretraining, it becomes easier to mitigate toxicity through post-training interventions. Additionally, the paper provides a theoretical justification for this phenomenon by introducing the concept of feature entanglement.

## update after rebuttal
Thank author for sharing new insights. No change are made. I am looking forward to "Role of Model Size in Optimal Toxic Data %" study in future work.

**Claims And Evidence:**

Claim in this paper is well supported by theoretical argument on feature entanglement. The model training result on Olmo-1B also supports author's claim.

**Essential References Not Discussed:**

Not that I am aware of.

**Experimental Designs Or Analyses:**

Experiment shows key evidence support the claim.
1. Experiment shows increasing proportion of toxic training data result in more toxic generation if no intervention applied.
2. When apply Inference-time Intervention, model trained using more toxic data result in lower toxicity aligned with authors' claim and theoretical analysis.
3. Experiment also shows with increasing toxicity data, model shows better MMLU performance and toxicity detection performance. With toxicity detection performance significantly improved at 25% toxicity data.
4. Author also compared multiple post training methods including Prompting, MEDA and INST, SFT, and DPO.

Critiques
1. The main concern is in Fig. 6, the decreasing trend on toxicity VS. toxic training data is not monotonic. Model performed the best at 10% with strong steering. Author doesn't explain why 10% shows the optimal performance, and there is no hypothesis on this.
2. Pretrain used clean + % of toxic data up to 25%. While increasing toxic data, clean data kept constant, therefore, the total amount of training token changed, would this impact model MMLU result in Fig. 4? Increasing training data size may reduce model's toxicity in general.

**Methods And Evaluation Criteria:**

- Author used MMLU, Toxigen, C4, and 4chan dataset. Those are commonly used data for LLM toxicity tasks. Those dataset well supports authors' claim and experiment

**Other Comments Or Suggestions:**

N/A

**Other Strengths And Weaknesses:**

- Provide sound theoretical justification and toy experiment to support feature entanglement argument.
- Experiment are quite comprehensive comparing

**Questions For Authors:**

1. Why do you use Olmo-1B model on this task instead of 7B based model such as Llama?
2. Would the model size be an important factor in determining the optimal percentage of toxic data?

**Relation To Broader Scientific Literature:**

Model alignment and safety has been widely studied in the past. Reducing generation toxicity while maintaining model's generation quality has always been a difficult task with wide industrial application. This paper leveraged feature entanglement framework build by Elhage
et al. (2022), and give clear theoretical justification on why including bad data is important to train a good model. This work has application in industry as well, proposing a novel approach on how to reduce model generation toxicity.

**Theoretical Claims:**

Author introduced theoretical proof on Entanglement of features. I manually verified this entanglement lower bound proof through frame theory. I also manually verified 2D and 3D space cases.

---

> ### Author Rebuttal · Authors · 2025-03-30
>
> We thank the reviewers for their thoughtful feedback and insightful questions; please see our point-by-point responses below.
>
> ## Non-Monotonic Toxicity Trend in Fig. 6
>
> We will add a discussion in L318 (right column) to address this:
> The initial decrease in toxicity is due to the model learning a more separable toxic representation, as toxicity is underrepresented in the original dataset. This disentanglement enables better detoxification via ITI and prompting. The later increase stems from overrepresentation of toxic features, which become entangled with underrepresented clean features again.
>
> ## Impact of Varying Token Counts on MMLU
>
> Figure 4 shows MMLU performance remains stable despite increased toxic data, suggesting the relevant knowledge is primarily derived from the clean data.
>
> ## Role of Model Size in Optimal Toxic Data %
>
> This is a valuable point. We believe the optimal toxic data amount depends on both model size and the nature of the data. A comprehensive study is needed to understand how data mixture scaling impacts performance, and we defer this to future work.

---

### Decision · Program_Chairs · 2025-05-01

**Decision:**

Accept (poster)

**Comment:**

The paper makes the surprising claim that having more toxic data in pre-training can actually
lead to better models because it makes it easier to align in post-training. The argument is that some amount of toxic data in the pre-training creates some disentanglement of features and that during post-training the LLM can more easily understand something is toxic and hence avoid producing it.

The reviewers pointed out as a limitation that similar flavors of this idea have appeared in prior work (Lee et al. Qi et al.) but the authors clearly state these inspirations and further explain that alignment is more effective when its not unlearning toxic generations but learns to bypass them- fitting in the proposed mechanism.

The main limitation is that the experiments are still somewhat preliminary and limited to 1B scale but this is understandable due to the cost and challenges involved, especially in pre-training. I still think that the paper makes a very intriguing claim and presents enough evidence to be accepted for publication.